# Nano Calcium Carbonate (CaCO_3_) as a Reliable, Durable, and Environment-Friendly Alternative to Diminishing Fly Ash

**DOI:** 10.3390/ma14133729

**Published:** 2021-07-02

**Authors:** Lochana Poudyal, Kushal Adhikari, Moon Won

**Affiliations:** Department of Civil, Environmental & Construction Engineering, Texas Tech University, Lubbock, TX 79409, USA; kushal.adhikari@ttu.edu (K.A.); moon.won@ttu.edu (M.W.)

**Keywords:** supplementary cementing materials (SCMs) and fly ash, nanotechnology and nano CaCO_3_, Portland limestone cement (PLC), ordinary Portland cement (OPC), workability and mechanical properties, performance and durability, environmental sustainability

## Abstract

Fly ash is widely used in the cement industry to improve the performance and durability of concrete. The future availability of fly ash, however, is a concern, as most countries are inclining towards renewable energy sources as opposed to fossil fuels. Additional concerns have been raised regarding the impact of strict environmental regulations on fly ash quality and variability. This paper, therefore, evaluates if nano calcium carbonate (nano CaCO_3_) can be used as an alternative to fly ash. This paper presents comprehensive testing results (fresh, hardened, and durability) for OPC (Ordinary Portland Cement) and PLC (Portland Limestone Cement) concretes with 1% nano CaCO_3_ and compares them to those for concretes with fly ash (both Class F and C). Compared to concretes with fly ash, OPC and PLC with nano CaCO_3_ presented improved testing results in most cases, including later age strength, permeability, and scaling resistance. As nanotechnology in concrete is a relatively new topic, more research on the efficient use of nanotechnology, such as for proper dispersion of nano CaCO_3_ in the concrete, has potential to offer increased benefits. Further, nano CaCO_3_ is environmentally and economically viable, as it has the potential to be produced within the cement plant while utilizing waste CO_2_ and generating economic revenue to the industry. Thus, nano CaCO_3_ has the potential to serve as an alternative to fly ash in all beneficial aspects—economic, environmental, and technical.

## 1. Introduction

Cement industries produce approximately 8 to 10% of global anthropogenic CO_2_ emissions [1]. The cement production is projected to exceed 6 billion metric tons by 2050 (from the current production of approximately 4 billion metric tons), adding more CO_2_ into the atmosphere [2]. Reducing the carbon footprint of the cement industry thus has been a global concern. The cement industry searches for ways to meet increasing demand while reducing the carbon footprint of the cement produced. The Cement Technology Roadmap has identified energy-efficient technologies, alternative fuels, clinker substitution, and emerging and innovative technologies as the four major paths for achieving environmental sustainability in the cement industry [3], and subsequent researches are underway [2,4,5,6,7,8]. One widely used approach, and probably the easiest and most economical method, is clinker substitution, where a proportion of cement is replaced by SCMs (Supplementary Cementing Material) such as fly ash, GGBFS (Ground Granulated Blast-Furnace Slag), silica fume, and limestone powder [9,10,11,12].

Fly ash has been used since the late 1930s [13] and is the most used SCMs so far. Fly ash is economical, environment-friendly, and easily available. Additionally, the use of fly ash in concrete provides improved performance and durability properties; fly ash specifically helps minimize ASR (Alkali Silica Reaction) as well as sulfate attack in concrete and meet the temperature requirement for mass placement [14,15]. Fly ash is a byproduct of burning pulverized coal in power plants and is pozzolan in nature. When mixed with cement, fly ash reacts with portlandite (calcium hydroxide) to produce calcium silicate hydrate, which contributes to increasing strength and reducing porosity, thus improving the durability of concrete.

Fly ash, however, may not be a sustainable and viable option in the years to come, as the future availability of fly ash and its quality are already a concern [16,17,18,19]. Most power plants across different nations are already making a switch towards natural gas or other renewable energy than coal. Additionally, the quality of fly ash has been declining due to the Environmental Protection Agency (EPA) regulations on limits of SOx, NOx, and mercury [20]. The use of ammonia in flue gas will help lower SOx and NOx emissions; however, this use will increase the concentration of ammonia in fly ash, leading to odor issues during construction. Likewise, activated carbon is injected into flue gas to absorb the mercury. Activated carbon in fly ash absorbs air-entraining admixture; thus, controlling adequate entrained air content in concrete becomes quite difficult and poses a real challenge to the concrete industry.

Further, to comply with the EPA regulations, power plants have started using river basin coal as their fuel source [18]. This change will reduce the supply of Class F fly ash, which is more effective than Class C fly ash in improving concrete durability as well as in controlling concrete temperature in mass concrete. Whether it be the reduced supply or the declining quality, the use of fly ash may not be a viable option in the years to come. Many research studies have been conducted to identify alternatives to fly ash [16,21,22]. The use of natural pozzolans such as pumice, metakaolin, rice husk ash, or natural zeolite is being investigated as an alternative to the use of fly ash, but these natural pozzolans also have long-term availability concerns because they are not distributed equally over the world [23].

Lately, nanotechnology has gained popularity where nanomaterials are being used as a partial replacement to cement clinker [24,25,26,27,28,29,30]. Nanomaterials change the structure of hydrated paste at a nano level, thereby dramatically improving both compressive and flexural strength, performance, and durability [24,25,26]. Several studies have concluded that incorporation of nanoparticles such as nano SiO_2_, nano TiO_2_, nano CaCO_3_, nano Fe_2_O_3_, nano Zr_2_O_3_, nano Al_2_O_3_, and nano graphene (CNTs and CNFs) in cementitious composites can significantly improve their performance and durability levels [26,31,32]. Most of these nanomaterials, however, come with high price, thus limiting their commercial implementation in the cement industry.

As compared to several others, nano CaCO_3_ is relatively cheaper [33]. Further studies have shown that nano CaCO_3_ has the potential to be produced within the cement plant while utilizing the waste CO_2_ from cement production [33,34]. A recent study by Batuecas et al. (2021) [34] shows that replacement of cement by 2% nano CaCO_3_ helps reduce the CO_2_ emission from cement plants by 69%. The CO_2_ emission from cement plant was reduced from 0.96 kg CO_2_ eq./kg cement to 0.3 kg CO_2_ eq./kg. Thus, the use of nano CaCO_3_ (despite lower replacement rates as compared to fly ash) helps meet two key benefits—economic and environmental—of using fly ash in concrete. This paper now evaluates the technical viability of using nano CaCO_3_ as compared to benefits offered by using fly ash in concrete.

Recent studies [24,35,36,37] have shown that the use of nano CaCO_3_ improved the early and later age strengths of Portland cement concretes. Additionally, a previous study by the authors presents comprehensive testing results on OPC and PLC with different replacement rates of nano CaCO_3_; 1% replacement provided the optimal performance, with increased later age strength and durability properties in both cement types. This paper compares the performance of OPC and PLC concretes with 1% nano CaCO_3_ to that of concretes with fly ash (both F and C) and evaluates if nano CaCO_3_ can be used as an alternative to fly ash. To the authors’ knowledge, no prior efforts have been done to evaluate the applicability of nano CaCO_3_ as an alternative to fly ash.

## 2. Materials and Evaluation Methods

### 2.1. Materials

Type I/II—Ordinary Portland Cement (OPC) with Blaine fineness of 390 m^2^/kg and type IL—Portland Limestone Cement (PLC) with Blaine fineness of 450 m^2^/kg were used as binders. Type IL PLC was produced by inter-grinding type I cement clinker with 15% of limestone in the cement plant. Table 1 shows the chemical composition of OPC (Lafarge, Ravena, NY, USA), PLC (Capitol Aggregates Inc., Austin, TX, USA), Class F fly ash (Boral Material Technologies, San Antonio, TX, USA), and Class C fly ash (Lubbock, TX, USA). Limestone with a nominal maximum aggregate size of 19 mm (0.75 in) and siliceous sand with a fineness modulus of 2.6 were used as coarse and fine aggregates (locally available), respectively. The gradation of coarse aggregate is shown in Figure 1. We used 98% pure white precipitated nano calcium carbonate (CaCO_3_) with an average diameter of 40 nm and surface area greater than 40 m^2^/g.

### 2.2. Evaluation Methods

This paper presents comprehensive testing results for four different concrete types—concretes with (1) OPC (and with 1% nano CaCO_3_), (2) PLC (and with 1% nano CaCO_3_), (3) OPC (replaced with 20% Class C fly ash), and (4) OPC (replaced with 20% Class F fly ash). The properties evaluated were: (a) slump and setting of fresh concrete, (b) strength and modulus of elasticity of hardened concrete, and (c) concrete durability properties—permeability, alkali silica reaction (ASR), and scaling resistance. In addition, SEM (Scanning Electron Microscope) images for different samples were obtained. The mixture proportions for 0.76 m^3^ (1 cy) of concrete are given in Table 2. The water-to-binder ratio was fixed at 0.47, with a total binder content of 295 kg per 0.76 m^3^ (650 lbs per 1cy) of concrete. No chemical admixtures were used in this experiment to limit the interference of other chemicals on the effects of nano CaCO_3_. During the mix preparation, nano CaCO_3_ and cement were placed in an electrically driven mechanical mixer and blended under high speed for three to four minutes prior to introducing them into the mixture. Previous studies [38], however, have shown that the addition of superplasticizers helps the proper dispersion of nanomaterials and enhances the performance.

Slump and setting tests were conducted as per ASTM C143 and ASTM C403, respectively.

For compressive strength and elastic modulus, cylindrical concrete specimens of 100 × 200 mm (4 in diameter and 8 in length) were tested at 3, 7, 28, and 56 days. The specimens were fabricated and moist cured in accordance with ASTM C192. A Rapid Chloride Penetration Test (RCPT) was conducted in accordance with ASTM C1202 using the Giatec Scientific test setup, where the charges passed through the specimen were automatically calculated at the end of six hours. For this testing, cylindrical specimens, 100 × 200 mm (4 × 8 in), were cured for 56 days as per ASTM C192. The cured specimens were then cut to the proper size (100 × 50 mm (4 × 2-in)) for testing, as prescribed by ASTM C1202. An Alkali–Silica Reaction (ASR) test was performed on prismatic specimens 25 × 25 × 254 mm (1 × 1 × 10-in), as per ASTM C1260, where sand with a high reactivity from a local source was used. The changes in length of the specimens were measured as per ASTM C157 at 5, 10, and 14 days. Scaling resistance was evaluated in accordance with ASTM C672. The testing was performed on beam specimens 150 × 150 × 610 mm (6 × 6 × 24-in) with plastic dikes of 19 mm (0.75 in) of height for the chloride solution. The specimens for SEM images were prepared as per ASTM C305, and the images were obtained using a Zeiss Crossbeam 540 FIB-SEM.

The properties of OPC and PLC concretes with 1% cement replacement by nano CaCO_3_ were compared with those of concretes containing fly ash in order to assess whether nano CaCO_3_ could fill in fly ash as an effective mineral admixture. All the testing results presented in this paper are the averages of those of three specimens, except for scaling resistance (single specimen). Most of the measured values were within one standard deviation (1-SD), with few data extending over 2-SD. For better understanding of the test results, all the figures (as appropriate) are provided with error bars representing 1-SD.

### 2.3. Statistical Analaysis

Tukey multiple comparison test (also referred to as Tukey HSD test) was adopted as a statistical tool in evaluating significant differences. Here, the test results for concretes with 1% nano CaCO_3_ were compared with the corresponding test results for concretes with fly ash. The Tukey test was used as it has a low false positive rate, i.e., it provides a higher level of confidence for the differences to be real.

## 3. Results and Discussions

### 3.1. Concrete Slump

Figure 2 shows the slump for different concrete mixes. The slumps for both class F and C fly ash concretes are relatively higher as compared to those for OPC concrete. Class C fly ash concrete has approximately 50% higher slump than the OPC; this difference can be attributed to the morphologic effect, resulting from the micro beads present in the fly ash acting as a lubricating agent in fresh concrete [39]. However, the PLC concrete had comparable slump values to those of fly ash concretes. It was also observed that the replacement of 1% nano CaCO_3_ had minimal effects on the slump for OPC and PLC concretes. Although the water demand is expected to increase with the use of finer particles, lower replacement levels did not seem to affect the workability [24].

### 3.2. Setting

As we replaced the cementitious materials with pozzolans such as fly ash, the pozzolans reacted with calcium hydroxide, a product of cement hydration, thus inhibiting early hydration. This had a pronounced effect on the set time of the concrete, as illustrated in Figure 3, with fly ash concretes displaying a larger set time compared to OPC and PLC concretes. Further, Class C fly ash concrete showed a lower set time compared to Class F fly ash concrete due to a higher amount of calcium oxide content in Class C fly ash.

The addition of nano CaCO_3_ further reduced the initial and final set times for both OPC and PLC concretes, with a larger reduction in PLC concrete. In PLC concrete, the combined dilution and filler effects increased the rate of hydration at early ages, thus lowering the set time. Additionally, the increased nucleation effect resulting in an increased C_3_S hydration rate also helped lower the set time for PLC [24].

### 3.3. Compressive Strength

The strength of concrete mainly depends on the amount of hydration products, the porosity of concrete, and the packing of the microstructure. Figure 4 illustrates the compressive strength achieved for all the concretes. In general, fly ash concretes have lower early strength compared to OPC and PLC. As shown in Figure 4, Class F fly ash concrete showed the lowest strength among all concrete types at 3 days and 7 days. Similar results were observed for testing at 28 days and 56 days, while studies have shown that the Class F fly ash concrete strength normally exceeds the OPC concrete strength at about 90 days [39,40]. It should also be noted that the pozzolanic reaction takes place even after years of concrete placement. On the other hand, the strength of Class C fly ash was comparable to that of OPC at 28 days and exceeded the corresponding values for OPC and PLC at 56 days.

Despite improved later age strength, one of the major limitations of using fly ash is the reduced early-age strength; this reduction limits its applicability, especially in cold weather regions. However, the use of nano CaCO_3_ improved the strength of both OPC and PLC concretes at all ages, while exceeding the corresponding strengths when compared to concretes with fly ash. The improved strength was the result of an enhanced nucleation effect, which increased the rate of hydration. In addition, the formation of extra hydration products such as carboaluminates improved the porosity and packing of the microstructure, thus increasing the later age strength [24]. Thus, as far as strength is concerned, nano CaCO_3_ serves as a better alternative to fly ash.

### 3.4. Elastic Modulus

The elastic modulus of concrete mainly depends upon the aggregate properties and porosity or densification of the paste. Since the same aggregate was used in all concrete types, no significant difference was observed. However, there seemed to be loose correlation between compressive strength and elastic modulus due to the denseness of the hydrated cement paste. Figure 5 shows the elastic modulus of all concretes. At all ages, fly ash concretes (especially Class F) displayed a relatively lower modulus value as compared to OPC and PLC concretes. In contrast to the reduced modulus of concrete with fly ash, nano CaCO_3_ improved the modulus values for both OPC and PLC. Nano CaCO_3_ did not compromise the development of concrete stiffness, in contrast to as fly ash, and could serve as a better alternative to fly ash.

### 3.5. Rapid Chloride Penetration Test (RCPT)

The RCPT measures concrete resistivity, not permeability; however, it has been shown that a correlation exists between concrete resistivity and permeability [41]. Figure 6 shows the charges passed in coulombs for all the concretes. It can be observed that the concretes with fly ash showed lower charges passed compared to OPC and PLC. The lower charges observed in fly ash concrete can be attributed to the formation of more CSH gel due to the pozzolanic reaction at later ages and increased density of the concrete. Class F fly ash concrete showed a lower value than Class C, as the amount of silica present in Class F was higher, leading to increased pozzolanic reaction in concrete.

On the contrary, PLC had the highest charges passed, followed by OPC concrete. The higher charge numbers passed in PLC concrete indicated dominant dilution effects in PLC. Further, as cited in Bonavetti et al. [42], chloride ions in the solution react with monocarboaluminate in PLC to form chloroaluminates, capturing the chloride ions and increasing the resistivity.

The addition of nano CaCO_3_, however, improved the resistivity of both OPC and PLC concretes. OP-1 concrete exhibited the highest resistivity among all concrete types; the charges passed in OP-1 were approximately 30% and 35% less than those in concretes with Class F and Class C fly ash. The significant improvement in OPC was due to the improved packing and enhanced microstructure with a greater amount of hydration products. Likewise, similar improvements, but at a lower scale, were observed for PL-1 concrete. PL-1 concrete showed comparable resistivity values to those of fly ash concretes. This shows the potential of improving the microstructure of concrete at later stages using nano CaCO_3_ instead of fly ash, thus improving the durability of concrete.

### 3.6. Scaling Resistance

Figure 7 presents the results for scaling resistance of concretes, where mass loss after every five cycles of freezing and thawing can be observed. For all concrete types, the rate of mass loss gradually increased in early cycles, followed by steady and decreased mass loss at later cycles. Class F fly ash concrete was observed to have the highest mass loss, followed by OPC and Class C fly ash. PLC experienced the least mass loss. Several factors, including porosity or strength, w/b ratio, method of curing, air entrainment, and type of coarse aggregates used, affect the scaling resistance of concrete [43,44]. Among these factors, air entrainment and porosity are considered the two major critical factors. In this test, concrete was mixed without the use of any air-entraining admixtures.

The addition of nano CaCO_3_ improved the scaling resistance of both OPC and PLC concretes. The OP-1 concrete showed a lower mass loss than both classes of fly ash, while PL-1 exhibited the least mass loss among all concretes. One of the reasons for the lowest mass loss in PL-1 could be the consumption of C_3_A at early ages to form carboaluminates, as explained by the inverse relationship between the amount of C_3_A and the scaling resistance of concrete [45].

Figure 8 shows the concrete surface of all samples after 20 freeze-and-thaw cycles with a calcium chloride solution. The fly ash concrete had maximum scaling, as seen in Figure 8, as all of the top layers were scaled off. The organic matter and carbon content present in fly ash reduced the effects of the retainment of air pockets, thus reducing the scaling resistance [46]. Also, a lower strength of fly ash at the early ages has been considered as another factor for increased mass loss in fly ash concrete [46]. This phenomenon can also be observed in the differences between mass losses in Class C and Class F fly ash concretes. Class C fly ash concrete showed relatively higher early strength than Class F fly ash; therefore, it had a smaller mass loss compared to Class F fly ash. It should also be noted that ASTM C672 is considered a very harsh laboratory testing and does not replicate the actual field condition. Case studies have shown that adequate air entrainment with lower fly ash content seems to have performed satisfactorily in the field [47].

Concretes with fly ash, thus, have lower resistance to scaling, which has been a concern in freeze-thaw environments. On the contrary, the use of nano CaCO_3_ improved the scaling resistance of both OPC and PLC concretes. As shown in Figure 8, visible reduction was observed in the amount of scaled concrete surface after the replacement of OPC and PLC with nano CaCO_3_. Thus, nano CaCO_3_ can help in mitigating another major limitation of fly ash concrete, thereby presenting a better alternative to fly ash in concrete applications in cold regions.

### 3.7. Alkali–Silica Reaction

The alkali–silica reaction (ASR) in concrete occurs when reactive silica present in the aggregates react with hydroxide ions in pore solution to form expansive silica gel, which could cause micro cracks in concrete when moisture is absorbed. The expansion of concrete due to ASR depends on several factors such as alkali content of the cement, amount and reactivity of silica present in aggregates, availability of moisture, and porosity of the concrete. Figure 9 presents the testing results from ASTM C1260. The fine aggregate selected in this evaluation is known for excessive ASR potential.

The maximum expansion was seen for the OPC concrete, and the least expansion was seen for the concrete with Class F fly ash. Several reasons for lower ASR in Class F fly ash concrete include the reduced porosity due to the pozzolanic reaction, binding of alkali in the CSH gel, and consumption of portlandite, which reduced the alkalinity of the pore solution. The expansion of Class C fly ash concrete was, however, much higher than that of Class F concrete, with an expansion nearly equal to that of OPC concrete. Research studies have shown that Class C fly ash does not reduce the expansion as effectively as Class F fly ash [14]. This difference has been attributed to the chemical composition of Class C fly ash; Class C fly ash has a higher lime content than Class F fly ash, thus contributing to the formation of portlandite rather than consuming it.

The addition of nano CaCO_3_ significantly reduced the expansion of OPC by approximately 45%. The reduced microstructure porosity due to the filling of the micropores of concrete could be one of the major reasons for this reduction in expansion, as explained by Lochana et al. [24]. Likewise, the expansion reduced by approximately 20% for PLC concrete. Thus, nano CaCO_3_ was effective in reducing the expansion of both OPC and PLC, with values lower than that of Class C fly ash. However, its ability to reduce concrete expansions was not as effective as for Class F fly ash. Where reducing the ASR potential is a major issue, other alternatives such as either a lower cement content or a cement with a low alkali content could be considered.

### 3.8. Statistical Analysis

Table 3 presents the summary of the Tukey HSD test analysis. Significant difference (*p* < 0.05) was observed in the performance of concretes with 1% nano CaCO_3_ as compared to concretes with fly ash. The compressive strength, at all ages, and the scaling resistance of both OP-1 and PL-1 were significantly higher compared to those of concretes with fly ash. The permeability of OP-1 was significantly reduced, while PL-1 had higher permeability when compared to concrete with fly ash (class C), but the difference was not significant. Likewise, the ASR expansion of both OP-1 and PL-1 was higher when compared to that of concrete with fly ash (class F), but the difference was not significant. Thus, the statistical analysis confirmed that nano CaCO_3_ outperformed fly ash in most of the tests, with few tests where fly ash showed relatively better results, which, however, were not significant. Further, confirming the previous study by the authors [24], concretes with 1% nano CaCO_3_ exhibited a significantly higher performance in all tests (except the ASR) when compared to traditional concretes without nano CaCO_3_.

### 3.9. Microstructure of Cement Pastes

SEM images of cement pastes were analyzed for further research. In Figure 10, the images on the left column (Figure 10a,c,e,g,i,k) correspond to 3-day hydration for F20, C20, OP-0, OP-1, PL-0, and PL-1, respectively, and the figures on the right column (Figure 10b,d,f,h,j,l) correspond to 28-day hydration. The 3-day hydration images for all cement paste show a fluffy structure with ettringite crystals, indicating early hydration. Similarly, the 28-day images for all samples show hardened CSH gel, portlandite crystals, and ettringite needles.

For fly ash cement paste at 3 days (Figure 10a,c), the ettringite crystals appeared smaller as compared to OP-0, indicating a slower hydration rate at early ages, as indicated by lower compressive strength and increased set time. The SEM images of 3-day hydration of Class F and Class C cement pastes look similar. However, at 28 days of hydration, Class C fly ash particles appeared more reactive than Class F fly ash particles. Class C fly ash particles appeared almost fully covered in CSH gel, as seen in Figure 10d. This reactivity could be one of the reasons for Class F fly ash concrete having lower strength than Class C fly ash concrete at 28 days.

For OPC and PLC at 3 days (Figure 10e,i), the images show ettringite crystals, CSH gel, and unhydrated products. After addition of nano CaCO_3_ (Figure 10g,k), the ettringite crystals seemed to be larger compared to those in OP-0 and PL-0, indicating a rapid increase in hydration rate at early ages that might have resulted in higher early compressive strength and reduced final set time. Additionally, calcium carboaluminates were confirmed in the images of PL-1, OP-1, and PL-0 cement pastes by EDS analysis, contributing to more hydration products. Thus, at 28 days of hydration (Figure 10h,l), the SEM images show a more compact structure compared to OP-0 and PL-0. This provided a denser structure, thus improving the permeability and later age strength of the concretes. This phenomenon could also explain reduced chloride permeability, ASR expansion, and improved scaling resistance of both concretes with nano CaCO_3_.

## 4. Summary and Future Recommendations

Fly ash has been widely used in the concrete industry to enhance concrete performance, while lowering concrete cost and mitigating adverse environmental impacts. The supply and quality of fly ash, however, have already been a concern in several regions of the world, and thus fly ash may not be a viable option for the years ahead. This paper evaluated if nano CaCO_3_ can be used as an alternative to fly ash. In this study, the performance of OPC and PLC concrete with 1% nano CaCO_3_ was compared with that of both Class F and C fly ash concrete; research consisted in the evaluation of fresh and hardened concrete properties, including mechanical properties and durability performance. The findings from this study can be summarized as follows:The concrete with nano CaCO_3_ had improved mechanical properties (compressive strength and elastic modulus) at all ages compared to fly ash concretes.The use of nano CaCO_3_ reduced the permeability of both OPC and PLC concretes. The permeability of OP-1 was lower than that of concretes with fly ash, while PL-1 had a permeability comparable to that of concretes with fly ash.The use of nano CaCO_3_ improved the scaling resistance of both OPC and PLC concretes, with the highest resistance for PLC concrete, thus mitigating the major limitation of concretes with fly ash for use in freeze–thaw environments.The addition of nano CaCO_3_ reduced the expansion of both OPC and PLC concretes by approximately 50% and 20%, respectively. The expansion of OPC and PLC concretes was lower than that of concrete with Class C fly ash but was not as effective as for Class F fly ash.SEM images showed that the microstructure of concrete improved with the addition of both nano CaCO_3_ and fly ash in concretes.

The testing results showed that both OP-1 and PL-1 (OPC and PLC concretes with 1% nano CaCO_3_) showed comparable performance, which in most cases, exceeded the performance of fly ash concretes. In addition to a higher performance, the use of nano CaCO_3_ can also be economical and more sustainable, as this concrete has the potential to be produced within the cement plant with the utilization of waste CO_2_. Thus, nano CaCO_3_ has the potential to serve as a reliable, economical, and environment-friendly alternative to fly ash. Further, moving towards a more environment-friendly alternative, more studies on the durability of PLC with nano CaCO_3_ should be initiated. As PLC with nano CaCO_3_ showed comparable or better results than OPC, there might be the possibility to produce PLC with higher percentage of limestone powder, thus reducing the cement clinkers.

## Figures and Tables

**Figure 1 materials-14-03729-f001:**
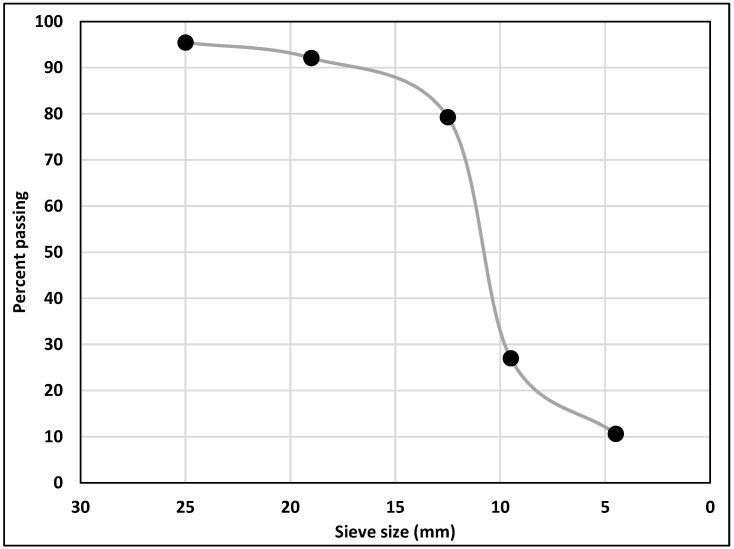
Gradation of Coarse Aggregate.

**Figure 2 materials-14-03729-f002:**
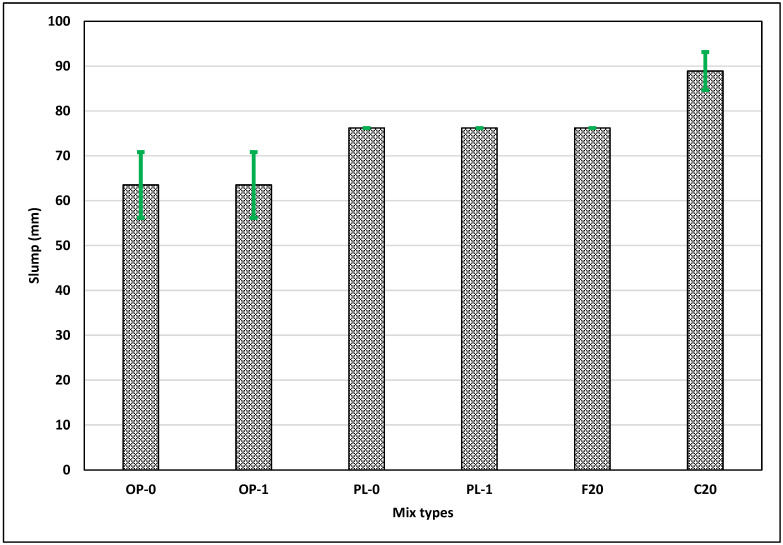
Results of the slump test for all mixes.

**Figure 3 materials-14-03729-f003:**
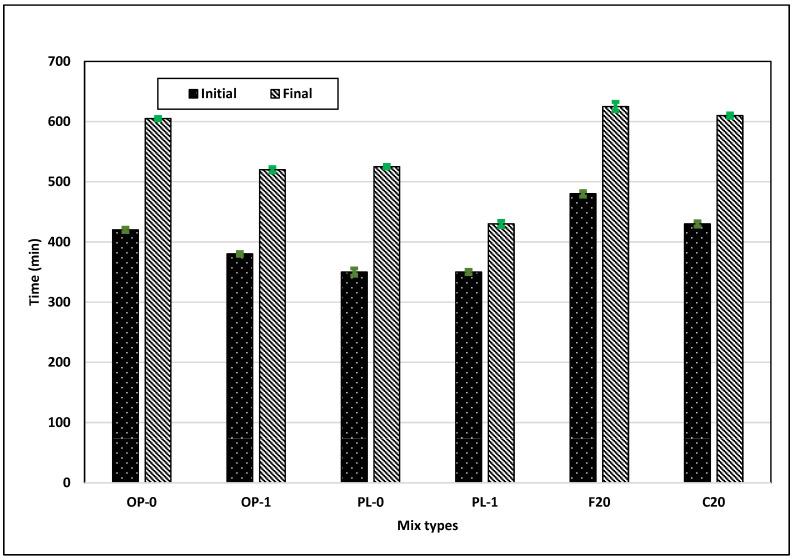
Results of the setting test for all mixes.

**Figure 4 materials-14-03729-f004:**
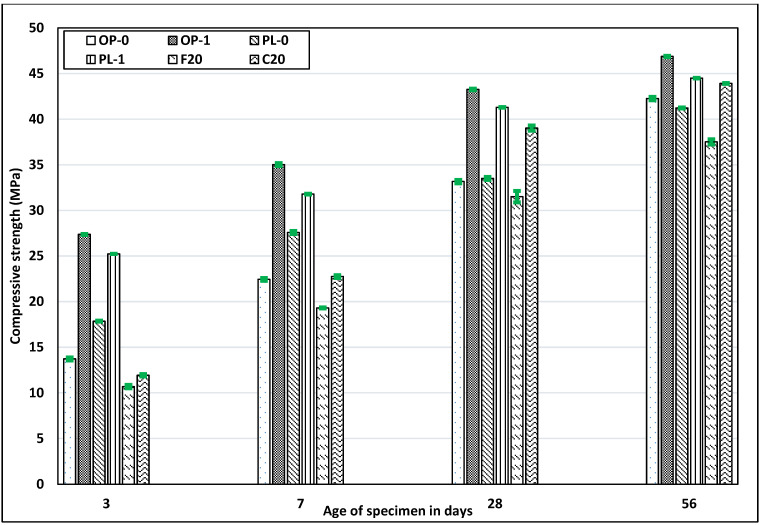
Results of the compressive strength test for all mixes.

**Figure 5 materials-14-03729-f005:**
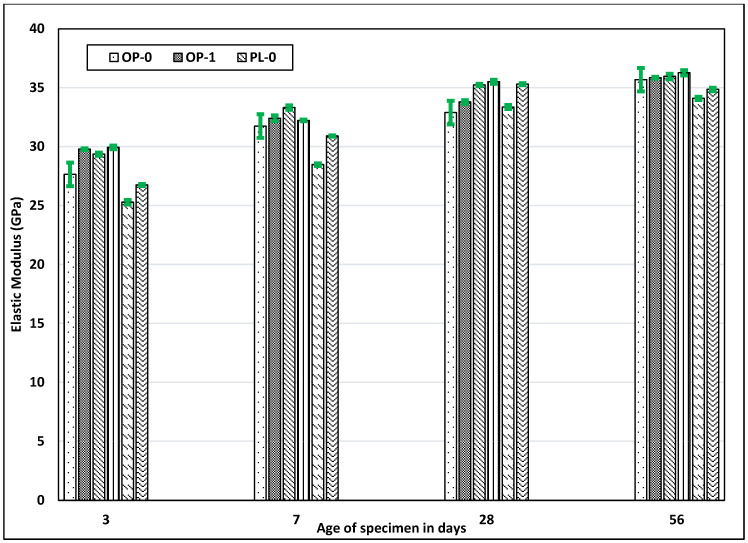
Results of the elastic modulus test for all mixes.

**Figure 6 materials-14-03729-f006:**
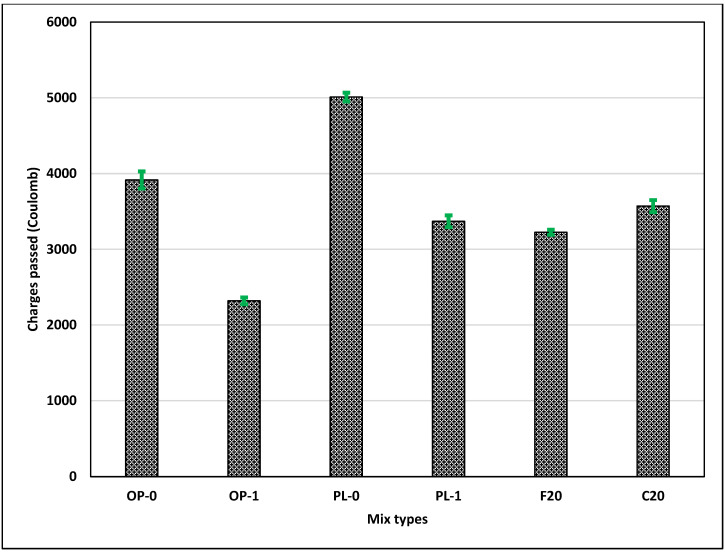
Results of the RCPT test.

**Figure 7 materials-14-03729-f007:**
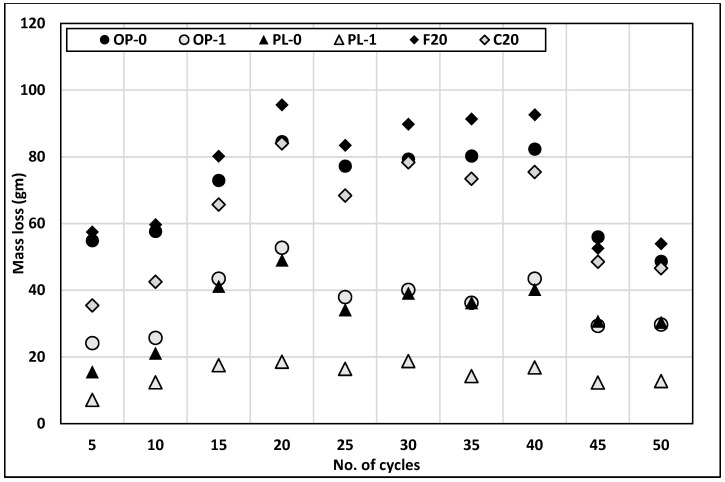
Results of scaling resistance.

**Figure 8 materials-14-03729-f008:**
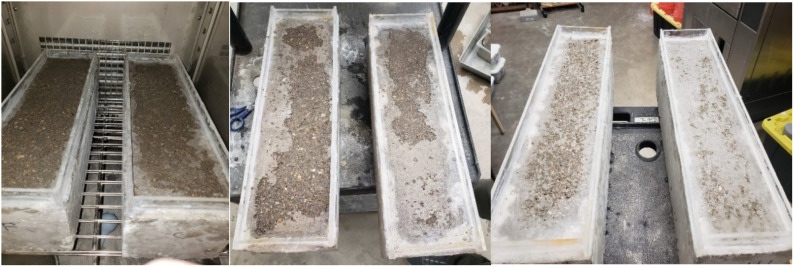
Scaling of concrete surfaces at 20 cycles (left to right: F20, C20, OP-0, OP-1, PL-0, PL-1).

**Figure 9 materials-14-03729-f009:**
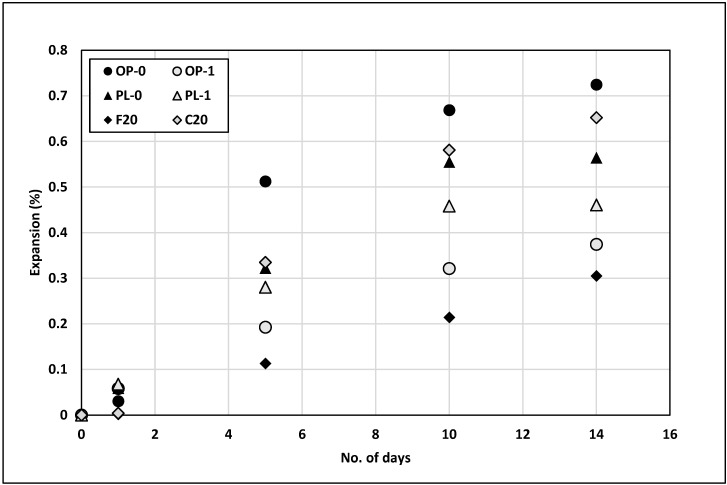
Results of the ASR test.

**Figure 10 materials-14-03729-f010:**
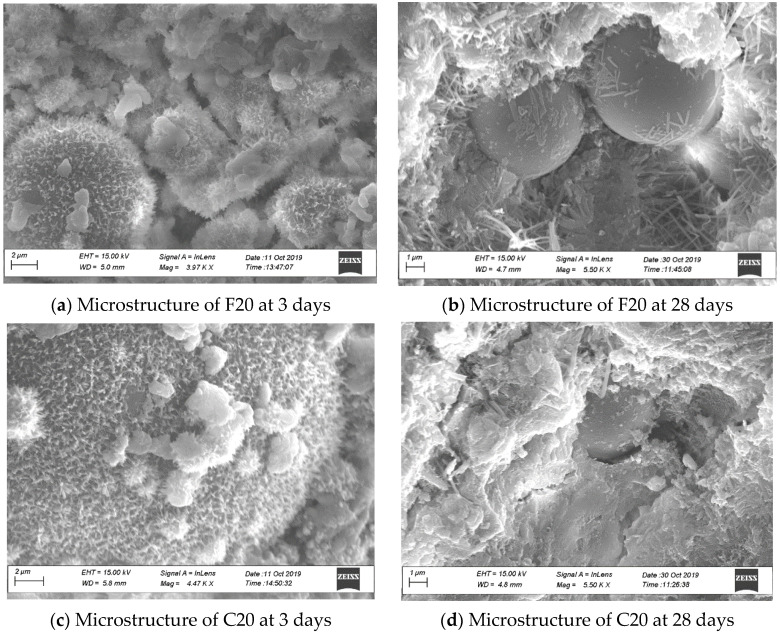
SEM images of the samples.

**Table 1 materials-14-03729-t001:** Chemical Compositions of Ordinary Portland Cement (OPC), Portland Limestone Cement (PLC), and Fly ash (% by weight).

Constituents	OPC Type I/II	PLC Type IL	Class C Fly Ash	Class F Fly Ash
SiO_2_	19.7	20.2	35.5	63.67
Al_2_O_3_	4.7	5.5	18.3	24.26
Fe_2_O_3_	3.0	1.8	7.4	5.08
CaO	62.1	65	25.9	2.70
MgO	3.7	1.2	5.3	0.93
SO_3_	2.9	3.8	1.4	0.25
Equivalent alkalis	0.59	0.38	2.15	-
LOI (Loss on Ignition)	2.29	6.1	0.2	3.21

**Table 2 materials-14-03729-t002:** Mix design for 0.76 m^3^ cy of concrete.

Mix Designation	Description	Nano CaCO_3_	Fly Ash	Water	Cement	Coarse Aggregates	Fine Aggregates
OP-0	OPC with 0% nano CaCO_3_	0	0	138	295	784	525
OP-1	OPC with 1% nano CaCO_3_	3	0	138	292	784	525
PL-0	PLC with 0% nano CaCO_3_	0	0	138	295	784	525
PL-1	PLC with 1% nano CaCO_3_	3	0	138	292	784	525
F20	OPC with 20% Class F fly ash	0	59	138	236	784	525
C20	OPC with 20% Class C fly ash	0	59	138	236	784	525

w/c ratio = 0.47 for all mixes, all units are in kg.

**Table 3 materials-14-03729-t003:** Summary of the Tukey HSD Analysis.

Sample	OP-0	PL-0	F20	C20	Test
OP-1	Y, >	Y, >	Y, >	Y, >	3 days Comp. Strength
PL-1	Y, >	Y, >	Y, >	Y, >	
OP-1	Y, >	Y, >	Y, >	Y, >	7 days Comp. Strength
PL-1	Y, >	Y, >	Y, >	Y, >	
OP-1	Y, >	Y, >	Y, >	Y, >	28 days Comp. Strength
PL-1	Y, >	Y, >	Y, >	Y, >	
OP-1	Y, >	Y, >	Y, >	Y, >	56 days Comp. Strength
PL-1	Y, >	Y, >	Y, >	N, >	
OP-1	Y, <	Y, <	Y, <	Y, <	Rapid Chloride Penetration
PL-1	Y, >	Y, <	N, <	N, >	
OP-1	N, <	N, <	N, >	N, <	Alkali Silica Reaction
PL-1	N, <	N, <	N, >	N, <	
OP-1	Y, <	N, >	Y, <	Y, <	Scaling Resistance
PL-1	Y, <	Y, <	Y, <	Y, <	

Y represents significant differences (*p* < 0.05), and N represents the opposite; > means higher than the values of concretes with fly ash; < means lower than the values of concretes with fly ash.

## Data Availability

The data that support the findings of this study are available from the corresponding author upon reasonable request.

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
