# Peer review of "Nano Calcium Carbonate (CaCO3) as a Reliable, Durable, and Environment-Friendly Alternative to Diminishing Fly Ash"

_materials, 2021, doi:10.3390/ma14133729_

Round 1
Reviewer 1 Report
Dear Authors,
The paper Nano calcium carbonate (CaCO3) as a reliable, durable, and environment-friendly alternative to diminishing fly ash by Lochana Poudyal, Kushal Adhikari and Moon Won is well suited for journal Materials. The authors of this article analyzed the results of the studies on the replacement of fly ash with nano calcium carbonate. The paper is interesting and scientifically valuable, especially due to the environmental impact.
The paper contains parts in good order: introduction, materials and methods, results, conclusions. Abstract highlights the problem in a good way and discusses the research presented in the article. Keywords – no comments. Here it should be noted that the word "Keywords" itself was entered in a font that is over standard height. Introduction – the problem from the technical and environmental side is well described, previous research is discussed and the research presented in the article is outlined. In the main part of the article, the authors discuss the materials, mixtures and dimensions of the tested samples. The results of individual studies were discussed. The main achievements of the article have been bulleted in the summary. The bibliography contains 49 items, including 2 from journal Materials.
The article was written enough well in English, is understandable for a reviewer, a person who does not speak English as a mother tongue.
Major shortcomings:
The number of samples, 3 specimens in most tests, is not too much. The obtained results should be commented on in terms of homogeneity. Taking the average of similar results is good, but if one of the results differs significantly from the others, it is a problem. There is no basis to analyze which results are more reliable. Such information would give readers an idea of the stability of the results (in terms of these 3 specimens).
Minor shortcomings:
- It is a pity that the version of the article for review does not contain line numbers.
- Page 3, Fig. 1 – the signature should be centered on the line and formatted like next.
- Page 8, Fig. 5 – in the picture instead of "Gpa" should be "GPa".
- Pages 12-14, Fig. 10 – in several cases, the scale in the photos is different, probably the authors did not have photos of the same scale; this should be noted in future research.
Author Response
Dear Editor and Reviewers,
Thank you very much for your suggestions on the paper. We found the comments very helpful, and we have tried our best to address all the comments while improving the quality of the paper. The revised manuscript contains point-by-point responses to the reviewer's comment provided. The response to the reviewer’s comment is marked in GREEN and associated changes in the manuscript are marked in BLUE. Please find the responses below for all the comments.

Reviewer 2 Report
The paper is quite well written, there are just a few minor points - few typos, non-uniform formatt of references.
However: there is a significant content overlap with another paper by the same authors: Mechanical and Durability Properties of Portland Limestone
Cement (PLC) Incorporated with Nano Calcium, Carbonate (CaCO3), Materials, Volume 14, Issue 4, Pages 1 - 192 February 2021 Article number 905, DOI 10.3390/ma14040905.
This includes large number of identical data, some completely identical figures.
Plus, in some cases SI units are not used at all - e.g. Figure 1, Table 2.
Author Response

(The authors gave the same response as above.)

Reviewer 3 Report
This paper explores the potential of nanocalcium carbonate (nano CaCO3) as an alternative to fly ash, for improving mechanical and durability properties of concrete. This research is based on a comprehensive test program (mechanical tests, Chloride penetration tests, scaling resistance, alkali-silica reaction) that was carried out on various concrete formulations: OPC and PLC concrete, the same concrete including 1% of nanoCaCO3, and OPC concrete with 20% replacement by fly ash of class C or F. Results shows that the concrete formulation including 1% nanoCACO3 shows superior mechanical properties and durability over the other materials, and nanoCaCO3 may be considered as a possible alternative to fly ash.
Globally, the paper is well written and experiments were carried out in a rigorous way. Many interesting results are presented which can be useful to researchers. It is therefore recommended to publish the paper after a minor revision. This latter is intended to add some missing information or minor inputs.
Requested corrections/modifications:
- Details of the product brands and manufacturers should be provided
- As stated in the introduction, a major objective of clinker substitution by fly ash is the reduction of the environmental footprint (reduced CO2 emission). In the present study, it is shown that nano CaCO3 should be used at very low content in the concrete formulation, and therefore their environmental impact in term of clinker substitution is very questionable. This should be argued in the text.
- In figure 1, the sieve size is expressed in inches. This unit must be converted in mm as most readers use the International system of units
- In section 2.2. and in Table 2, values expressed in cy and lbs units must also be replaced by their counterparts in the international system of units.
- In the caption of Table 1, it should be specified that compositions are in weight %.
- In Figure 7, in the title of the y axis, “mass” should be replaced by “mass loss”. What is the meaning of gms ?
Author Response

(The authors gave the same response as above.)

Round 2
Reviewer 2 Report
The paper is ready for publishing in its present form.